# How Do Post-Translational Modifications Influence the Pathomechanistic Landscape of Huntington’s Disease? A Comprehensive Review

**DOI:** 10.3390/ijms21124282

**Published:** 2020-06-16

**Authors:** Beata Lontay, Andrea Kiss, László Virág, Krisztina Tar

**Affiliations:** 1Department of Medical Chemistry, Faculty of Medicine, University of Debrecen, H-4032 Debrecen, Hungary; lontay@med.unideb.hu (B.L.); kissand@med.unideb.hu (A.K.); lvirag@med.unideb.hu (L.V.); 2MTA-DE Cell Biology and Signaling Research Group, Faculty of Medicine, University of Debrecen, H-4032 Debrecen, Hungary

**Keywords:** neurodegenerative Huntington’s disease, misfolded proteins and aggregates, impaired cellular events, posttranslational modifications

## Abstract

Huntington’s disease (HD) is an autosomal dominant inherited neurodegenerative disorder characterized by the loss of motor control and cognitive ability, which eventually leads to death. The mutant huntingtin protein (HTT) exhibits an expansion of a polyglutamine repeat. The mechanism of pathogenesis is still not fully characterized; however, evidence suggests that post-translational modifications (PTMs) of HTT and upstream and downstream proteins of neuronal signaling pathways are involved. The determination and characterization of PTMs are essential to understand the mechanisms at work in HD, to define possible therapeutic targets better, and to challenge the scientific community to develop new approaches and methods. The discovery and characterization of a panoply of PTMs in HTT aggregation and cellular events in HD will bring us closer to understanding how the expression of mutant polyglutamine-containing HTT affects cellular homeostasis that leads to the perturbation of cell functions, neurotoxicity, and finally, cell death. Hence, here we review the current knowledge on recently identified PTMs of HD-related proteins and their pathophysiological relevance in the formation of abnormal protein aggregates, proteolytic dysfunction, and alterations of mitochondrial and metabolic pathways, neuroinflammatory regulation, excitotoxicity, and abnormal regulation of gene expression.

## 1. Introduction to Huntington’s Disease

Protein misfolding diseases such as Alzheimer’s disease (AD), Parkinson’s disease (PD), Huntington’s disease (HD), and other prion diseases are disorders of the central nervous system (CNS) associated with protein aggregation and toxicity. In all of these diseases, acquisition of an abnormal secondary structure by specific proteins accompanies aggregation, and this structure subsequently propagates in the brain in a prion-like manner [1]. At the cellular level, these neurodegenerative diseases are associated with the disturbance of normal cellular processes, including protein translation [2,3], post-translational modification (PTM), protein degradation [4], and mitochondria homeostasis [5]. These abnormal cellular processes are all hallmarks of misfolded protein stress. Accumulation of incorrectly folded proteins is at the root of these incurable neurodegenerative diseases.

HD is a progressive, lethal, neurodegenerative hereditary disorder characterized by the adult-onset of motor dysfunctions, psychiatric disturbances, intellectual decline, dementia, and finally death [6]. Intraneuronal inclusions are distinctive features of the disease. HD is caused by the mutation of the CAG repeat in the huntingtin (*Htt)* gene, which results in a faulty protein product with an expanded (>36) polyglutamine (polyQ) stretch. 

The wild-type huntingtin protein (wtHTT) is ubiquitously expressed in many cell types and localized to different cellular compartments. wtHTT was described to interact with a panoply of proteins and to act as a scaffold for various types of autophagy. Hence, HTT is important in intracellular protein clearance [7]. HTT is also involved in a wide variety of cellular functions, such as cell signaling, apoptosis, and transcriptional regulation. 

Expansion of the polyQ repeat in the mutant huntingtin protein (mHTT) makes the protein prone to misfolding [8]. Monomers of mHTT are organized into large aggregates or different types of oligomers with different levels of toxicity. In terms of histopathology, HD is mainly characterized by the intranuclear accumulation of HTT-derived aggregates in the brain cortex and striatum of patients [9] and by neuronal dysfunction. This eventually results in the loss of spiny neurons in the striatum and subsequent neuronal damage and cell death.

HD pathogenesis is a very complex process. Looking at the cellular level, HD comprises protein function associated with, but not limited to, transcription, proteolysis, mitochondrial homeostasis, cytoskeleton alteration, and neuroinflammation. Molecular and cellular interactions are extremely diverse. As such, the presence of mHTT leads to the destabilization of transcriptome, proteome, and metabolome, disrupting a large number of cellular processes. 

Post-translational modifications (PTMs) have a very crucial role in the regulation of HTT function. PTMs involve the chemical modification of a protein. These enzymatic or non-enzymatic reactions include the covalent attachment of a simple chemical group to the protein as in phosphorylation, acetylation, and methylation; the addition of complex groups like in glycosylation or lipid modifications; or the addition of a group and proteolytic cleavage of the target protein at a specific amino acid residue in case of ubiquitination and proteolysis, respectively. PTMs can be reversible or non-reversible processes, and they regulate protein function by targeting specific subcellular compartments, interacting with ligands or other proteins, or by bringing about a change in their functional state including catalytic activity or signaling. PTMs are tightly regulated and aberrant PTMs that often result in pathological conditions. 

Post-translational modifications of the wtHTT protein seem to play a major role in the subcellular localization and the regulation of protein–protein interactions. Dysregulated PTMs in the pathogenesis of HD may lead to the susceptibility of proteins to aggregate. Moreover, PTMs affect the elements of neuronal signaling pathways even in the presymptomatic stage of HD, independently of the formation and presence of abnormal HTT aggregates. A recent study systematically summarized and identified PTMs of HTT, which may act as potential modulators of HD proteinopathy [10]. On the other hand, as a cause or consequence, dysregulated PTMs of specific proteins (listed in Table 1) also underlie the molecular and cellular pathogenesis of HD, perturbing normal cellular homeostasis and function. In this review, we try to give a systematic overview of recent findings focused on the principal PTMs, highlighting their roles in the pathomechanism of HD.

## 2. Post-Translational Modifications in Selected Cellular Events of the Diverse Pathomechanism of Huntington’s Disease 

### 2.1. PTMs in Abnormal HTT Protein Aggregation

HTT is a large, 348 kDa protein. The disease-causing polyQ stretch is found within exon-1 encoding the N-terminal fragment of the protein (HTTex1), which is preceded by 17 amino acids at the N-terminal (N17) and is followed by the ~40 residue proline-rich domain (PRD) [68]. The N17 sequence functions as a nuclear export signal and its PTMs are thought to modulate subcellular localization and the clearance of mHTT. The PRD has a function in HTT aggregation, protein–protein recognition, and mHTT turnover [69]. Based on structural studies, N17 is predominantly disordered in solution but adopts an α-helical conformation in amyloid fibrils [70,71]. The PRD includes a polyproline helix, while the polyQ region adopts multiple conformations (including α helix, random coil, and extended loop) which are influenced by the flanking protein regions [70]. The rest of the protein is poorly characterized but contains several HEAT repeats that are important in protein–protein interactions. 

HTT is subject to various types of PTMs, and these may influence the toxicity, aggregation propensity, and proteolytic degradation of mHTT. 

Ser/Thr-specific phosphorylation has a high impact on the aggregation properties of HTT. Within the N17 region, phosphorylation at the Thr-3 [72], Ser-13, and Ser-16 [73] residues attenuate mHTT aggregation and toxicity. The Thr-3 phosphorylation level is dependent on polyQ-length [72], and it is markedly reduced in HD patients [74]. This phosphorylation inhibits the aggregation properties of mHTT by reverting the conformational changes caused by the polyQ stretch and stabilizing the α-helical structure of the N17 region [74,75,76]. Protein phosphatase 1 (PP1) was identified as a modulator of HTT aggregation via dephosphorylation of phospho-Thr-3 [77]. Recently, single-molecule counting technology was developed to quantify HTT phosphorylation at Thr-3 [78]. Phosphorylation of Ser-13 and/or Ser-16 inhibits fibril formation. It also promotes in vitro internalization and nuclear targeting of HTTex1 aggregates by disrupting the amphipathic α-helix in the N17 region [79,80]. Ser-13 and Ser-16 phosphorylation also promote HTT clearance by proteasomes and lysosomes. Casein kinase 2 (CK2) [80] and the inflammatory kinase, IκB kinase (IKK) [81,82] were shown to catalyze the addition of phosphate groups to these sites. N6-furfuryladenine, a molecule produced during oxidative DNA damage, is used by CK2 to phosphorylate N17 and, thus, potentiate the elimination of mHTT inclusions [83]. Administration of the ganglioside, GM1, can also correct for hypophosphorylation in the N17 region [84]. IKK-induced phosphorylation of HTT also activates other PTMs of adjacent lysine residues, including acetylation, ubiquitination, and SUMOylation, which enhances HTT clearance by lysosomes and proteasomes [82]. 

Several other phosphorylation sites were identified outside the N17 region using mass spectrometry and/or site-directed mutagenesis. Among those, Ser-116 is highly effective in reducing HTT toxicity [85]. Nemo-like kinase (NLK) interacts with mHTT and phosphorylates it at Ser-120, thereby promoting the ubiquitination and degradation of mHTT via the proteasome pathway. NLK levels are decreased significantly in HD human brain and animal models. Importantly, overexpression of NLK in the striatum attenuates brain atrophy. NLK preserves levels of the striatal dopamine- and cAMP-regulated phosphoprotein of 32 kDa (DARPP32), resulting in reduced mHTT aggregation in HD mice [86]. Ser-421 of mHTT can be phosphorylated via the insulin-like growth factor (IGF-1) pathway by Akt and serum- and glucocorticoid-induced kinase (SGK). This phosphorylation can also protect against polyQ-induced toxicity [87,88]. Phosphorylation of Ser-421 regulates the direction of axonal transport in neurons [89,90] and reduces the nuclear accumulation and cleavage of HTT [91]. The Ser/Thr-specific protein phosphatases, PP1, PP2A [92], and calcineurin (PP2B) [93], were implicated in the dephosphorylation of phospho-Ser-421. Inhibition of these phosphatases is protective against neuronal cell death. Phosphorylation of HTT at Ser-434 [94], Ser-1181, and Ser-1201 [95] by cyclin-dependent kinase 5 (CDK5) has also been reported to be protective. STriatal-Enriched protein tyrosine Phosphatase (STEP) deletion reduces the size of mHTT aggregates by an undescribed molecular mechanism [96].

Acetylation of all the lysine residues within the N17 region (Lys-6, Lys-9, Lys-15) suppresses the formation of aggregated fibrils and inhibits HTT-lipid interactions [12]. However, acetylation of Lys-6 (but not Lys-9 or Lys-15) switches off the effect of phosphorylated Thr-3, indicating crosstalk of PMTs in the regulation of HTT properties [75]. Acetylation of Lys-444 improves the clearance of mHTT by targeting it to autophagosomes and enhancing mHTT degradation. This residue is acetylated by the acetyltransferase CREB-binding protein (CBP) and deacetylated by HDAC1 [97]. Three other acetyl-lysine sites (Lys-178, Lys-236, and Lys-345) were identified that might modulate HTT proteolysis or its association with lipids [98]. Mass spectrometry analysis and site-directed mutagenesis studies revealed that phosphorylation and acetylation sites that can modulate mHTT toxicity are located in clusters within protease-sensitive domains throughout the full-length HTT [11,99]. 

The G553E SNP of human *HTT* alters post-translational myristoylation of HTT causing pathogenic proteolysis of the protein [16].

As opposed to phosphorylation, differences in ubiquitination between wtHTT and mHTT are less studied. Ubiquitination sites were recently identified in soluble and insoluble fractions of brain lysates from the Q175 knock-in mouse model for HD and compared to ubiquitination of wild-type Q20. The main endogenous sites for ubiquitination of soluble HTT are Lys-6, Lys-9, Lys-132, Lys-804, and Lys-837. Wild-type HTT was mainly ubiquitinated at Lys-132, Lys-804, and Lys-837, while ubiquitination of mHTT was reduced in the soluble fraction and was ubiquitinated at Lys-6 and Lys-9. Insoluble fractions of both wtHTT and mHTT showed ubiquitination at Lys-6 and Lys-9. Moreover, increased ubiquitination and Lys-48 polyubiquitin linkages were identified in the insoluble fraction [13].

SUMO (small ubiquitin-like modifier) proteins are conjugated to proteins as a PTM. Eukaryotes express at least one member of the SUMO family. SUMOylation, similar to ubiquitination, is essential to many cellular processes, including transcriptional control, DNA repair, and regulation of protein–protein interactions, subcellular localization, and proteolytic processes [100,101,102]. In humans, the SUMO family members include SUMO1, SUMO2, and SUMO3. Recently, a detailed and comprehensive SUMO proteomics study was published combining all human SUMO proteomics data, which provided evidence of many SUMOylated proteins and SUMOylation sites [103,104]. Recent publications support the importance of SUMOylation in neurodegenerative disorders. Therefore, we opted to summarize recent data on how SUMO modifiers are involved in HD [102,105]. Earlier studies by *Steffan* et al. suggest that increased SUMOylation of HTT facilitates the disassembly of aggregates, leading to soluble monomer HTT formation and surprisingly increased neurotoxicity [14]. A study using flow cytometry for pulse-shape analysis (PulSA) in neuroblastoma Neuro2 cells analyzed transcriptional signatures before and after inclusion assembly. The data showed reduced SUMO2 pathways in both stages compared to control with the normal length of HTT, indicating an adaptive response of SUMO pathways to help the sequestration of HTTex1.

In recent years, one of the most studied E3 SUMO ligases was the HTT-selective PIAS1. PIAS1 regulates HTT accumulation and SUMO modification in cells [106]. An earlier publication demonstrated that PIAS1 reduction prevented the HD-associated phenotype and the accumulation of insoluble mHTT, improving synaptic health in an R6/2 mouse model. The authors suggest that PIAS1 may link protein homeostasis and neuroinflammation by modulating the formation of mHTT. Thus, PIAS1 is a potential therapeutic target for treating HD [107].

A recent bioinformatics analysis and research study identified a Ras homolog in the striatum as a potential target for HD therapy. The Ras Homolog Enriched in Striatum (RHES) was characterized as a small GTP-binding protein with SUMO-E3 ligase activity. RHES SUMOylates mHTT along with other proteins, but not wtHTT or ataxin-3, another polyQ containing protein. SUMOylated mHTT escapes insoluble aggregate formation and exhibits neurotoxic activity. However, SUMOylation of mHTT by RHES can be prevented by blocking RHES farnesylation at Cys-263, which is required for membrane localization of RHES. Mutating the conserved Cys-263 of RHES abolishes the SUMOylation, disaggregation, and cytotoxicity of mHTT. However, mutation of Ser-33, which is required for the GTPase activity of RHES, does not prevent the SUMOylation and cytotoxic activity of mHTT, suggesting that the SUMOylation and cytotoxicity of mHTT are not associated with GTPase activity. RHES is also involved in the SUMO-mediated vesicle transport of mHTT [19,20]. 

A study recently demonstrated that three lysine residues (Lys-6, Lys-9, and Lys-15) within the N17 of HTT could be SUMOylated. The SUMOylation of these lysines inhibited fibril formation and promoted the formation of larger SDS soluble aggregates. Moreover, SUMOylation of HTT inhibited HTT–lipid interactions and caused steric hindrance to the binding interaction. Although SUMOylation of mHTT prevents lipid interaction, which leads to membrane damage, SUMOylation of mHTT promotes other toxic mechanisms [108]. 

Normal HTT is palmitoylated at Cys-214 by huntingtin interacting protein 14 (HIP14) and the HIP14-like protein (HIP14 L). A palmitoylation resistant mutation at this site leads to defects in HTT trafficking and subcellular localization and promotes inclusion formation. Moreover, the polyQ expansion reduces the palmitoylation of wtHTT [15]. The functions of HIP14 and HTT are interdependent. wtHTT serves not only as a palmitoylation substrate but also modulates the palmitoylation of HIP14. HIP14 activity requires the palmitoylation of wtHTT, and HIP14 activity correlates with palmitoylation levels. wtHTT promotes HIP14 palmitoylation and, subsequently, the modification of HIP14 substrates such as SNAP25 and GluR [109].

### 2.2. Disrupted Proteolytic Pathways: PTMs in Abnormal HTT Protein Degradation

Eukaryotic cells contain two major types of proteolytic machinery: the lysosomal proteases and the 26S proteasome [110]. The 26S proteasome is a complex oligomeric structure that is present in both the cytoplasm and the nucleus of eukaryotic cells. The 26S proteasome is an ATP dependent protease composed of the 20S, which contains the proteolytic active sites, and the 19S regulatory particle [111]. The 19S regulatory particle is required for polyubiquitinated substrate recognition, unfolding, translocation, and deubiquitinating of substrates, and it is primarily directed by six distinct ATPases. The 26S proteasome generally requires a Lys-48 linked polyubiquitin chain conjugated to the substrate protein. In humans, more than 1000 proteins are involved in ubiquitination by creating, recognizing, and removing lysine-linked polyubiquitin chains. Ubiquitination of the substrate protein requires three enzymatic steps, including ubiquitin activation by E1 enzymes, ubiquitin conjugation by E2 enzymes, and ubiquitin ligation by E3 ligases.

The ubiquitin-proteasome machinery (UPS) is involved in the regulation of mHTT aggregation and toxicity. The proteasome can eliminate and reduce mHTT. Downregulation of proteasome activity promotes the aggregate formation of mHTT in both cell and animal models of HD [112,113,114] and the accumulation of toxic HTT protein results in further disruption of proteasome activity. Furthermore, the lack of efficient ubiquitination of mHTT also leads to inefficient proteasomal degradation and, subsequently, intracellular aggregates. Recently, Juenemann and colleagues demonstrated that inclusion bodies dynamically recruit ubiquitin and enzymes that are catalytically active in both ubiquitination and deubiquitination processes. However, the exact role of these enzymes in inclusion bodies is still to be explored [115]. 

One shared characteristic of many neurodegenerative diseases is the disruption of protein homeostasis. Research targeting neurodegenerative diseases has mainly focused on specific proteins; therefore, many studies focused on substrate-specific E2 and E3 enzymes. However, recent studies highlighted the role of the E1 ubiquitin-activating enzyme, UBA1, in neuronal homeostasis and neurodegeneration. Mammalian cells express only two E1 ubiquitin-activating enzymes, UBA1 and UBA6 [116]. UBA1 is highly conserved with two existing isoforms, UBA1a and UBA1b, and it is involved in the regulation of the cell cycle status of neurons, neurotransmitter release, and axon pruning [117]. The role of UBA1 has mainly been studied in relation to the pathogenesis of spinal muscular atrophy (SMA). However, in vitro studies revealed that the inhibition of the ubiquitin-activating enzyme E1 promotes the formation of high molecular weight mHTT in unaffected tissues, such as cerebral and peripheral tissue extracts. In addition, the level of the enzyme declines with age, suggesting that the impaired degradation of mHTT by the proteasome occurs via the ubiquitin-activating enzyme E1 leading to the accumulation of toxic HTT [118]. 

In HTT, the N17 and the adjacent polyQ stretch are highly disordered. The mHTT aggregates contain the N-terminal part of the HTT, ubiquitin, proteasomal components, and many other proteins. The ubiquitin-conjugating enzyme, E2 W (Ube2W), is the only known ubiquitin-conjugating E2 enzyme that preferably initiates ubiquitination on the N-termini of proteins with unstructured regions. Ube2W affects the level of soluble and intermediate species of mHTT but does not alter the level of aggregated protein. A deficiency in Ube2W E2 ligase leads to an increased level of soluble mHTT and decreased cellular toxicity in cultured cells and a mouse model of HD. Without direct evidence of how Ube2W alters the solubility of mHTT, Bo Wang and colleagues offered several hypotheses: (a) the N-terminal ubiquitination of mHTT, which stabilizes the protein, leads to elevated aggregate formation; (b) ubiquitination triggers other PTMs on the protein leading to aggregate formation; or (c) Ube2W indirectly regulates toxic aggregate formation through the ubiquitination of SUMO-2, which is already conjugated to HTT [119].

E3 ligases also determine degradation of mHTT and HD pathology. Numerous E3 ligases can recognize misfolded proteins with polyQs, enhancing the ubiquitination of mHTT and promoting their proteasomal degradation. The E3 ligases include Ube3a, Skp1-Cul1-F-box, Parkin, and the Hsp70-interacting protein, CHIP [120,121]. In this review, we focused on the most recent findings on the involvement of different E3 ligases in HD pathological processes.

Misfolded proteins and aggregate formation can induce endoplasmic reticulum (ER) stress leading to neuronal dysfunction and apoptosis. Homocysteine-induced endoplasmic reticulum protein (Herp) is an E3 ligase and early marker for ER stress. Immunofluorescent and co-immunoprecipitation studies showed that overexpressed Herp and HTTex1 with 160 polyQ interact in N2 a cells. Furthermore, Herp enhances the proteosomal degradation of soluble mHTT by promoting its ubiquitination and suppressing the formation of mHTT aggregates. However, the upregulation of Herp expression can reduce the level of mHTT but is unable to clear it out entirely. Thus, the authors speculate that upregulated Herp can only delay the progression of HD and not prevent it [122].

Misfolded proteins preferentially accumulate and cause neurotoxicity in neurons rather than glia cells in neurodegenerative disorders. One explanation for this phenomenon is that the less vulnerable astrocytes in HD have different Hsp70-interacting protein (CHIP) E3 ligase activity. CHIP is a co-chaperone of Hsp70 that binds to the chaperone-substrate complex and promotes Lys-48-linked polyubiquitination of the misfolded protein when it is mono-ubiquitinated. This leads to proteasomal degradation in astrocytes. Another co-chaperone of Hsp70, HspBP1 inhibits the E3 ligase activity of CHIP. *Zhao* and colleagues show that HspBP1 is more abundant in neurons than in astrocytes at both the transcriptional and protein level. Their findings offer a mechanistic insight into why astrocytes can clear accumulated mHTT more efficiently than neurons. They suggest that inhibiting HspBP1 promotes the clearance of accumulated mHTT and might provide an alternative approach to treat HD [123].

Most of the E3 ligases target mHTT via Lys-48 polyubiquitination in ubiquitin to promote its degradation by the proteasome. However, a recent study [124] demonstrated that mHTT is ubiquitinated through the atypical position at Lys-63 in ubiquitin by the E3 ligase, WWP1. In turn, Lys-63 ubiquitination positively regulates the level, aggregation, and toxicity of mHTT. In an HD mouse model, the authors showed increased expression of WWP1, which enhanced mHTT stability and made mHTT more prone to aggregation due to the atypical ubiquitination at Lys-63.

With a recently engineered bispecific antibody, which recognizes Lys-11/Lys-48-linked ubiquitin chains, *Yau* and colleagues identified mHTT aggregates with 73 polyQ that were decorated strongly with Lys-11 and Lys-48-linked chains in differentiated neurons and brains in an HD mouse model. Pathological HTT species are processed by Lys-11/48-linked quality control. Based on this observation, the authors raised a new possibility of how aggregation-prone proteins accelerate the progression of neurodegenerative diseases. They overexpressed the 73Q-HTT to form Lys-11/48-linked aggregates and inhibited the proteasome to stabilize misfolded nascent polypeptides. They observed that Lys11/48-linked chains were redistributed from 73Q-HTT to nascent proteins diminishing the ability of cells to fight aggregation. The 73Q-HTT and nascent proteins compete for a limited pool of enzymes that are necessary for the Lys-11/48-linked quality control [101].

Protein quality control can also be regulated by linear ubiquitination in HD. In linear ubiquitination, the C-terminus of the ubiquitin can form a peptide bond with the N-terminal methionine of an acceptor ubiquitin, resulting in M1 ubiquitination. The HOIL-1-interacting protein (HOIP) is a component of the linear ubiquitination chain assembly complex (LUBAC), and the only known E3 ligase that is capable of assembling a linear ubiquitin chain. LUBAC is recruited to poly-glutamine aggregates of mHTT, and LUBAC recognizes the aggregates as a cellular pathogen. Thus, a novel role for LUBAC was described in protein quality control by decreasing proteotoxicity and promoting the removal of misfolded proteins [125].

Data are emerging, demonstrating the role of different deubiquitinases in regulating protein quality control in HD. Ubiquitination is not only tightly controlled by ubiquitin ligases but also by deubiquitinases. Deubiquitinases target the isopeptide bond between the lysine residue of the target protein and the C-terminal glycine of ubiquitin. One candidate deubiquitinase is the highly conserved YOD1 (also known as OTUB2) with a preference towards Lys-48 and Lys-63 linked ubiquitin. The YOD1 level is increased by mHTT toxicity. The elevated level of YOD1 ameliorated cytotoxicity and altered proteasomal activity in cultured cells, suggesting that YOD1-deubiquitinating activity affects the degradation of mHTT and can degrade HttQ74 regardless of whether ubiquitin is Lys-48 or Lys-63-linked [126]. Another deubiquitinase, Usp12, is also neuroprotective in HD via enhancing the autophagic degradation of mHTT in rodent- and patient-derived neurons. Interestingly, the neuroprotective function of Usp12 does not require its catalytic activity, suggesting a regulatory function independent of the deubiquitinating activity [127].

HD also modulates the function of autophagy [128,129,130,131,132], although certain studies suggest that the two protein clearance pathways are impaired differently in different tissue types before the onset of HD [133]. In addition, cross-talk between the proteasome and autophagy pathways seems to be affected in HD. Earlier evidence suggests that engulfment of organelles by autophagosomes is defective in HD, mainly due to aberrant interactions of p62, polyubiquitin chains, and mHTTm [134]. p62 works as a signaling hub in a variety of cellular events. As an autophagy adaptor, p62 recognizes and brings ubiquitinated proteins to autophagosomes for degradation. One study showed that impairment of the UPS leads to elevated levels of p62, suggesting the up-regulation of autophagy pathways [135]. However, the molecular mechanism of action remained unknown. Recently, it was demonstrated that p62 is upregulated in both healthy and HD cells upon proteasomal inhibition. Furthermore, different subcellular localizations of p62 were shown in HD cells compared to the control after inhibition of the proteasome. p62 bodies were bound to Lys-48-linked polyubiquitin aggregates, and targeted them for proteasomal degradation in healthy, but not in HD cells. This suggests that aberrant subcellular localization and positioning of p62 in HD cells reduce protein aggregate clearance by the proteasome [136].

Another axis that mediates cross-talk between the proteasome and autophagy is the Usp14-Hsc70 axis. Usp14 is a proteasome-bound deubiquitinating enzyme, highly expressed in the nervous system. Recent studies have also established a role for Usp14 in autophagy. Usp14 is a common denominator of proteasomal and autophagy degradation [137]. Moreover, Usp14 negatively regulates autophagy by suppressing the Lys-63-linked ubiquitination of beclin-1 [138]. Using proteomic and in vitro cellular approaches for HD, Srinivasan and colleagues demonstrated that Usp14 and the chaperon Hsc70 dynamically interact. This interaction is enhanced by the inhibition of the proteasome with functional consequences in HD for modulating autophagy and might provide an alternative approach for therapeutic target development [139].

Proteolytic cleavage of HTT has a major impact on the molecular pathogenesis of HD. HTT contains several proteolytic sites and is the substrate for a variety of proteases. Enhanced proteolysis in HD-affected neuronal cells leads to the release of N-terminal fragments with expanded polyQ sequences. The polyQ expansion leads to misfolding and self-assembly into amyloid-like fibrillary aggregates [140,141,142], which form intranuclear inclusions in neurons. According to recent studies, cytotoxicity is attributed mostly to polyQ-expanded mHTTex1 fibrils, rather than oligomers or misfolded monomers [143].

Caspase-6 (CASP6) is critically involved in synaptic function, plasticity, and neurite pruning [144]. CASP6 can cleave mHTT at Asp-586, and it is critical for the pathogenesis of HD in an HD mouse model. Recently, another cleavage site at Asp-572 was identified in mHTT. This cleavage is mediated by caspase-1. The inhibition of caspase-1 decreases the cleavage at Asp-586, suggesting that CASP6 is inhibited through caspase-1. Furthermore, the inhibition of caspase-1 decreases the aggregation of mHTT and increases the turnover of the soluble mHTT, suggesting a protective role of caspase-1 inhibition in HD [18].

CASP6 was also demonstrated to be palmitoylated by palmitoyl acetyltransferase HIP14 at Cys-264 and -277 residues. These modifications hampered the substrate binding and dimerization of CASP6 and led to its inactivation. Both a Hip14-/- palmitoylation deficient mutant and the YAC128 mouse model of HD displayed increased CASP6 activation [17], where HIP14 was dysfunctional. Reduced palmitoylation of CASP6 results in cleavage of HTT and the generation of mHTT fragments. This suggests that pharmacological inhibition of CASP6 activity might improve the HD phenotype.

The activity of caspase-9, an initiator caspase and key player in the intrinsic pathway, is regulated by the apoptotic protease-activating factor 1 (Apaf-1), a component of the apoptosome. The lack of caspase-9 activation has significant pathophysiological consequences leading to degenerative diseases. An earlier study showed increased Apaf-1 levels in HD brains [145]. Etoposide initiates the interaction of Apaf-1 with Cullin-4 B, which results in enhanced ubiquitination of Apaf-1 in HEK-293 cells. Ubiquitinated Apaf-1 is then able to bind to p62 and formed aggregates in the cytosol.

Moreover, the complex can activate caspase-9 under conditions where the proteasome is impaired. This suggests that ubiquitination of Apaf-1 might have a regulatory role in neurodegenerative diseases. However, the above-described mechanism still needs to be further elucidated in HD [21].

### 2.3. Tau Impairment and Aberrant Cytoskeleton in HD

The microtubule-associated protein, tau, is widely expressed in the central nervous system, particularly in neurons, where it localizes almost exclusively in the axons. Only small amounts of Tau can be found in the nucleus and dendrites. Tau plays a critical role in the assembly and stabilization of microtubules, thereby maintaining neuronal morphology and facilitating axonal transport [146]. Tau is an intrinsically disordered protein with four functional regions: the N-terminal region, the proline-rich domain, the microtubule-binding domain, and the C-terminal region. Tau has two distinct groups of isoforms referred to as tau-3R or tau-4R, based on the number of basic repeats (3 or 4) in the central microtubule-binding regions. Tau-4R isoforms have a higher affinity for microtubules. Thus, tau-4R promotes microtubule assembly more effectively. The binding of tau to microtubules is regulated by phosphorylation, but other PTMs, such as acetylation, methylation, glycosylation, ubiquitination, and degradation/cleavage are also involved [147]. Phosphorylation reduces the affinity of tau for microtubules, resulting in microtubule instability. Several kinases, such as glycogen synthase kinase 3β (GSK-3β), cyclin-dependent kinase 5 (CDK-5), cyclin-dependent kinase 1 (CDK-1), mitogen-activated protein kinase (p38), c-Jun N-terminal kinases (JNK), protein kinase A (PKA), protein kinase C (PKC), calmodulin kinase II (CaMK-II), and CK2 are implicated in phosphorylation of Tau [148]. Dephosphorylation of tau is catalyzed by PP1, PP2A, PP2B (calcineurin), and PP2C phosphatases. Aberrant hyperphosphorylation of tau favors its dissociation from microtubules. This promotes the self-association and formation of neurofibrillary tangles (NFT), the main neuropathological hallmark of tauopathies, such as Alzheimer’s disease [149]. Recent studies have demonstrated the presence of aggregated tau inclusions in HD brains, putting HD on the list of tauopathies. These tau deposits reside either in the cytoplasm of neurons (often in the form of perinuclear ring structures) or in the nucleus, where they form newly described rod-like structures, named tau nuclear indentation (TNIs) or tau nuclear rods (TNRs) [22]. Tau alterations observed in HD include increased total protein levels, the appearance of additional truncated forms of tau [22], a shift in tau-4R/3R isoform ratio toward an increase in 4 R isoform level [22], and tau hyperphosphorylation [150,151]. Altered PTMs may contribute to the development of tau pathology in HD. The imbalance of tau isoforms is produced by an increase in the total abundance and phosphorylation of SRSF6, a splicing factor that favors the production of 4R tau [22]. Previously, Dyrk1A (dual-specificity tyrosine phosphorylation-regulated kinase 1 A) was reported to phosphorylate SRSF6 in its proline-rich domain [23]. Increased phosphorylation of tau was detected at Ser-396/Ser-404, Ser-199, and Ser-202/Thr-205 epitopes, but not at Ser235 or Ser262/Ser356 [150,151,152,153]. Blum and co-workers proposed that mHTT induces tau hyperphosphorylation, subcellular redistribution, and aggregation through direct protein–protein interaction [150]. The activity of tau kinases in HD samples is not consistent with the observed tau hyperphosphorylation. A marked decrease in the expression and activity of the major tau kinase, GSK-3β, was observed in the striatum and cortex of HD patients [22,154], while GSK-3β activity was elevated in hippocampal samples [153]. In other HD models, unchanged or even reduced expression/activity of tau kinases was detected. On the other hand, the inactivation of tau phosphatases is a more probable cause of aberrant tau phosphorylation, since downregulation of PP1, PP2A, and PP2B was detected in different HD mouse models [24,150]. Indeed, the calcium channel blocker, lercanidipine, proved to have neuroprotective effects in 3-nitropropionic acid-induced HD rats via modulation of the Ca^2+^/PP2B/NFATc4 pathway [155]. In a recent study, elevated levels of caspase-2 and truncated tau species were detected in the brain of HD patients, indicating that caspase-2-mediated tau cleavage may play a role in the pathogenesis of HD [25].

Cytoskeletal abnormalities, including axonal transport dysfunctions and dendritic spine destabilization, also contribute to the pathophysiology of HD. Axonal transport occurs along microtubules with the aid of the motor proteins, kinesin, and dynein. Axonal transport is also modulated by microtubule-associated proteins (MAPs), such as MAP2 and tau [156]. Vesicular transport of brain-derived neurotrophic factor (BNDF) is altered in HD [157], but the underlying molecular mechanism has not been clearly described. Loss of normal HTT, direct blocking of cargo movement by mHTT aggregates, and sequestration of motor proteins into mHTT aggregates can all contribute to the transport deficits [158,159]. In addition, a decrease in α-tubulin acetylation has been associated with mHTT. Acetylation occurs on the Lys40 of α-tubulin, a residue localized on the luminal side of polymerized MTs. Acetylation levels are determined mainly by the opposing actions of α-tubulin acetyltransferase 1 (ATAT1) and histone deacetylase 6 (HDAC6) [160]. Pharmacological inhibition of HDAC6 in neuronal cell cultures ameliorated vesicular transport and increased the clearance of diffuse mHTT [26,27]. On the contrary, the genetic depletion of HDAC6 in mouse models not only failed to attenuate HD progression but also worsened some behavioral deficits, even though it effectively increased α-tubulin acetylation [161,162]. Inhibition of another histone deacetylase, HDAC4, was shown to slow down mHTT aggregate formation. HDAC4 regulates autophagy-mediated degradation of mHTT by deacetylating and destabilizing the microtubule-associated protein 1 S (MAPS1) [28].

The cause of degenerative changes at dendrites of HD-affected neurons has not been revealed yet [163]. *Brito* et al. reported that mHTT toxicity results in elevated CDK5 activity in the brains of HD mice. Elevated CDK5 activity disrupts the CDK5/DARPP32/β-adducin signaling pathway and leads to increased β-adducin phosphorylation at Ser-713. This modification prevents the interaction of β-adducin with actin filaments, leading to an unstable dendritic spine cytoskeleton, and contributes to the depressive-like phenotype in HD [29,164].

### 2.4. PTMs Associated with Mitochondrial Abnormalities and Defects in Energy Metabolism in HD

Strong evidence indicates that mitochondrial dysfunction plays a critical role in the pathogenesis of HD. For example, HD patients show excessive weight loss despite normal food intake [165]. Glucose metabolism is decreased in affected brain regions of HD patients, and post-mortem brain samples exhibit decreased mitochondrial respiratory complex activity [166,167]. Moreover, mitochondrial malfunction occurs not only in affected brain regions but also in peripheral tissues such as skin fibroblasts derived from HD patients [168]. Mitochondria, isolated from lymphocytes of HD patients, exhibit decreased Ca^2+^-buffering capacity and altered mitochondrial membrane potential [169]. In most eukaryotic cells, mitochondria form a dynamic network and are subject to continuous fission and fusion. Imbalanced mitochondrial dynamics are a crucial underlying mechanism for neurotoxicity in HD and might cause the HD-linked mitochondrial defects [170,171]. Mitochondrial fission, predominately regulated by the GTPase activity of dynamin-related protein 1 (Drp1), provides quality control for the organelle [172,173,174]. Drp1 plays a central role in the nervous system, and excessive Drp1 function may contribute to the pathological progression of neurodegenerative diseases by triggering excessive mitochondrial fragmentation [175,176]. mHTT physically interacts with mammalian Drp1 with a higher affinity than normal HTT and stimulates its GTPase activity to promote mitochondrial fission. Increased Drp1 activity and enhanced mitochondrial fragmentation eventually lead to neuronal cell death [177]. Many mechanistic details have been identified that regulate Drp1 function in mammalian cells. They can be classified into two main categories: PTMs and regulated turnover [178,179]. Several kinases and phosphatases are known to influence Drp1 activity and recruitment to the mitochondria [179]. In this review, we aim to focus on the most recent findings pertaining to the PTMs of Drp1 in HD summarized in Figure 1A.

Phosphorylation of Drp1 at Ser-40 and Ser-44 by GSK-3β promotes the GTPase activity of Drp1 and results in increased mitochondrial division and neuronal cell apoptosis. CDK5 triggers the over-activation of Drp1 by phosphorylating Ser-579 in HD mouse striatal cells. This phosphorylation is linked to mitochondrial dysfunction [30]. In addition, MAPK1 binds to and phosphorylates Drp1 at Ser-616. Phosphorylation at Ser-616 increases in HD knock-in mouse-derived striatal cells and is associated with elevated mitochondrial fragmentation [31,32]. Drp1 phosphorylation by PKA at Ser-656 decreases Drp1 activity, leading to decreased mitochondrial fission events and neuroprotection [180]. This effect is antagonized by calcineurin. Increased basal calcineurin activity in HD tissues causes dephosphorylation and hyperactivation of Drp1, increased mitochondrial translocation and activation, and ultimately, mitochondrial fragmentation [181]. In primary cultured neurons from the striatum of transgenic HD mouse and human post-mortem brain, elevated NO caused by the expression of mHTT led to S-nitrosylation of Drp1 (SNO-Drp1). SNO-Drp1 induced excessive mitochondrial fragmentation, synaptic damage, and neuronal loss, suggesting that SNO-Drp1 contributes to the pathogenesis of HD [33,34].

Another component orchestrating mitochondrial function and make-up is mitochondrial biogenesis. SIRT3 is one of the mitochondrial sirtuins that has a major role in regulating mitochondrial biogenesis via its deacetylating activity. In HD, SIRT3 can protect dopaminergic neurons by reducing ROS through deacetylating MnSOD [35,36]. Other substrates of SIRT3 described in HD include the Lon protease and the mitochondrial transcription factor A (TFAM). SIRT3 can activate the Lon protease by deacetylation, resulting in the degradation of oxidized aconitase. Activated Lon protease acts on TFAM and TFAM, and in turn, regulates mtDNA replication and transcription, thereby preventing cells from mitochondrial dysfunction originating from misfolded protein accumulation [35,37]. Furthermore, TFAM is a direct deacetylation substrate for SIRT3. SIRT3 deacetylates TFAM, and thus, enhances its expression [182] (Figure 1B).

HD is also associated with abnormal energy metabolism. Currently, there are three hypotheses for the metabolic dysfunction in HD based on experimental results gained from postmortem tissues and HD mouse and cell culture models. The first hypothesis proposes that the dysfunction of mitochondrial trafficking, dynamics, and biogenesis promotes the translocation of organelles at inappropriate synaptic sites [183]. The second hypothesis is that the impairment of energy generating complexes in mitochondria originates from the deficits of their expression and functionality [184]. The third hypothesis argues for a model where mHTT accumulation directly induces transcriptional and proteomic changes that lead to energetic dysfunction as the cells try to act by a compensatory homeostatic response [185].

This compensatory mechanism was described in astrocytes of human HD striatum. The constant low glucose concentration in each brain region in HD triggers the astrocytes to adapt by metabolically reprogramming their mitochondria to use endogenous, non-glycolytic metabolites such as fatty acids as an alternative fuel. The striatum is enriched in fatty acids, and mitochondria are reprogrammed to oxidize them as an energy source at the cost of generating reactive oxygen species. Toxicity occurs when oxygen damage exceeds the energy benefits of fatty acid oxidation [186]. The mitochondrial complex II-III activity is deregulated in the brain of HD patients resulting in abnormal mitochondrial depolarization, leading to the generation of free radicals and oxidative damage [187].

Decreased ATP production and evidence of increased oxidative stress were described in glycolysis, the Krebs cycle, and the electron transport chain in HD tissues. A strong correlation between the level of energy metabolism impairment and the increased number of CAG repeats of mHTT has been observed. The efficiency of conversion of glucose to ATP is maximal in the resting state in brains of HD patients. When the neuronal activity is increased, the energy-generating biochemical pathways response with an increase in ATP production failed [188]. Moreover, the disruption of glucose metabolism and normal mitochondrial function promoted neuronal death [189] and contributed to the pathogenesis of HD [190]. Glucose metabolism decreased significantly in HD-affected striatal and cortical tissues, and elevated lactate levels were observed in the cerebral cortex and basal ganglia [191].

Dysregulation of the canonical Wnt/β-catenin pathway modifies the metabolic enzymes in HD. Without activation, β-catenin is phosphorylated by casein kinase 1 (CK1) and glycogen synthase kinase 3 in the cytoplasmic β-catenin destruction complex. An E3 ubiquitin ligase recognizes the phosphorylated β-catenin and promotes its ubiquitination and proteasomal degradation. Wnt/β-catenin activation by binding to the Wnt ligand results in the recruitment of the destruction complex to the membrane. This leads to hampering of the destruction complex, and finally, the stabilization of β-catenin. The accumulated β-catenin enters the nucleus and activates the transcription of target genes. Upregulation of the Wnt/β-catenin pathway induces aerobic glycolysis, named the Warburg effect, through activation of the glucose transporter, pyruvate kinase M2, pyruvate dehydrogenase kinase 1, monocarboxylate lactate transporter 1, and lactate dehydrogenase kinase-A and inactivation of the pyruvate dehydrogenase complex. This energy harvesting process is less efficient in terms of ATP production compared to oxidative phosphorylation, because of the TCA cycle shunt [38].

Insulin and insulin-like growth factor 1 (IGF-1) are homologous growth factors that bind to the insulin receptor (IR) and IGF-1 receptor (IGF-1 R), respectively. Ligand-receptor binding causes the subsequent activation of phosphoinositide 3-kinase (PI3K) or insulin receptor substrates promoting the activation of the pro-growth Akt, mTOR, and ERK/MAPK pathways. These steps lead to the phosphorylation of transcription factors such as nuclear factor–κB (NFkB) and forkhead box O1 (FOXO). The deregulation of these intracellular signaling pathways is associated with reduced Akt and ERK activation and a decrease in HTT phosphorylation in the HD brain. Insulin and, more profoundly, IGF-1 induces the activation of these pathways, resulting in the inhibition of the GSK-3β and FOXO1 signaling pathways [192]. Above all, insulin stimulates neuronal glucose uptake and its conversion to pyruvate and the restoration of the intracellular ATP level. Moreover, both insulin and IGF-1 and insulin-stimulated HTT phosphorylation at Ser-421 in the HD cells lead not only to the restoration of neuronal metabolism but also to neuroprotection. IGF-1 and insulin also rescue energy levels in HD peripheral cells, indicated by increased ATP and phosphocreatine and decreased lactate levels. IGF-1 effectively amends O_2_ consumption and mitochondrial membrane potential in HD [193]. 

### 2.5. Cell Death: Focus on Excitotoxicity

Neurodegeneration in HD seems to be selective for medium-sized spiny neurons (MSN) composing approximately 90% of striatal neurons [194] and for their projections to the areas of substantia nigra and globus pallidus of the basal ganglia. Striatal neurons receive glutamatergic input from several sources, including the cortex and thalamus and, in return, they stimulate glutamate receptors on striatal MSNs. Excitotoxic neuronal death and striatal vulnerability in HD may be caused by increased glutamate release from cortical afferents, reduced uptake of glutamate by glia cells, hypersensitivity of post-synaptic N-methyl D-aspartate (NMDA) receptors (NMDARs), as well as other receptors, or impaired downstream signaling of glutamate receptors [195,196,197]. Several lines of evidence support the regulatory role of PTMs in the pathomechanism of glutamate-mediated neuronal death, namely excitotoxicity in HD [198], as summarized in Figure 2.

#### 2.5.1. NMDA Receptor: Function and Regulation by PTMs in HD

NMDARs are ionotropic glutamate receptors that require the binding of the glutamate ligand to open [199] and membrane depolarization to remove the receptor blocking the Mg^2+^ ion [200]. The activation of NMDAR triggers an influx of Ca^2+^, which activates numerous signal transduction pathways [201]. Overstimulation of NMDAR lies in the background of neurodegenerative diseases [202], causing the pathological conditions of excitotoxicity leading to apoptosis and necrosis [203].

The primary NMDA receptor-mediated neuronal cell death pathway is parthanatos. The central mediator of this regulated necrotic cell death route is the nuclear DNA nick sensor enzyme poly(ADP-ribose) polymerase-1 (PARP1). Protein poly(ADP-ribosyl)ation (PARylation) by PARP1 is initiated by DNA damage and involves the transfer of multiple ADP-ribose moieties from NAD^+^ to acceptor proteins [204]. Similar to HTT [205], the poly(ADP-ribose) polymer serves as a scaffold, integrating the recruitment of DNA repair effector proteins [206]. Information on the potential role of PARPs or PARylation in HD is scarce. Vis et al. demonstrated a strong expression of PARP1 in neurons and glia cells in the HD caudate nucleus by immunohistochemistry [207]. Increased PARP1 immunoreactivity indicates a possible role for PARP1 in HD. The PARP inhibitor, INO-1001, attenuates neurological dysfunction in the R6/2 mouse model of HD [208]. Suppression of striatal atrophy, neuronal intranuclear inclusions, microglial reactions, and protected morphology of striatal neurons are evidence for the effect of the PARP inhibitor. Since PARP is a central mediator of excitotoxicity [209,210], it is plausible that inhibition of PARP may inhibit HD at least in part via blocking excitotoxicity. Indeed, a recent study found that the weak PARP inhibitor 3-aminobenzamide either alone or especially in combination with the NMDA receptor antagonist, memantine, improved bioenergetics (striatal NAD and ATP content) and mitochondrial markers (striatal succinate dehydrogenase activity) in the 3-nitropropionic acid-induced model of HD [211]. These findings prove that PARP activation contributes to the pathomechanism of HD. The role of PARP1 in HD likely involves the mediation of excitotoxic cell death via parthanatos, but the enzyme may also be linked to HTT aggregation. The PAR polymer may enhance the liquid-liquid phase separation of neurodegeneration-associated hnRNP A1 and TDP-43, and thus, regulate the dynamics of disease-associated protein complexes [212]. Whether these PAR-dependent phase separation events also occur in HD requires further investigation.

The internalization of the GluN2B subunit of NMDAR from the plasma membrane upon synaptic activity is a coordinated work of several protein kinases and phosphatases. The GluN2B subunit is phosphorylated on Tyr-1472 by Fyn/Src kinases that are associated with the subunits of NMDA receptors through scaffold proteins, including the postsynaptic density protein 95 (PSD-95). Phosphorylation prevents endocytosis of the receptor, and therefore, increases its surface expression [213,214]. NR2 A and NR2 B subunits of NMDA are also phosphorylated by Src kinases at Tyr-837 and Tyr-842 residues, respectively. This results in a reduction of endocytosis, and consequently, the stabilization of NMDAR on the synaptic surface [44]. WtHTT, but not the mutant HTT protein increases the phosphorylation level, and thus, the activity and targeting of phospho-Src and PSD-95 to the membrane fraction in these cells [45]. Increased Src-related tyrosine phosphorylation of NMDARs sensitizes neurons to excitotoxic stimuli. However, neither early symptomatic R6/2 mice nor R6/2 transgenic mice showed striatal NMDAR-mediated excitotoxicity compared to their wild-type littermates [215]. Another protein kinase, the Ca^2+^ -activated non-receptor tyrosine kinase, Pyk2, also plays a vital role in the regulation of NMDA receptor function by recruiting Src kinases and PSD-95 to the receptor. Pyk2 activity is decreased in patients with HD, and its deficit contributes to hippocampal impairments in the mouse model of HD [216].

In contrast to Src/Fyn, the phosphorylation of GluN2B on Ser-1480 by CK2 disrupts anchoring with the postsynaptic density and allows NMDARs to diffuse laterally to extrasynaptic sites. In addition, the calmodulin-dependent protein kinase II (CaMKII)/PKC tandem regulates the synaptic expression of NMDARs by phosphorylating the GluN2B on Ser-1303 residue [217].

Phosphorylation can regulate the surface expression of the receptor by regulating endocytosis and by affecting protein export from the ER to the plasma membrane. NMDAR is also phosphorylated at serine/threonine residues by PKC and PKA [218]. These modifications trigger NMDA-induced currents and increase NMDAR surface expression and activity [219]. Consistent with these data, decreased levels of the mRNA for the PKCβ isoform in the striatum of symptomatic R6/2 mice correlated with the lack of sensitivity to NMDAR excitotoxicity. PKA also phosphorylates NMDAR at Ser-897 of the NR1 subunit [218]. Together with PKC phosphorylation at Ser-896, this phosphorylation regulates the trafficking and exit of assembled NMDAR from the ER [220]. Parallel with this evidence, striatal tissue from symptomatic N171-82 Q mice showed enhanced phosphorylation of NR1 at Ser-897 in striato-nigral and striato-pallidal MSNs relative to controls [221].

Another key player in the downregulation of synaptic NMDAR expression is the membrane-associated phosphatase, STEP. It dephosphorylates the regulatory tyrosine residue (Tyr-1472) in GluN2B, inducing the lateral transport of NMDAR to extrasynaptic sites [222]. Pre-symptomatic enhancement of the activity of a synaptic-specific STEP61 isoform was detected in an HD mouse model. STEP61 was shown to dephosphorylate both the GluN2B subunit of NMDAR and ERK1 MAPK, causing the hypersensitivity of the receptor and neuronal cell death, respectively [43].

Calcineurin is also an inevitable modulator of the phosphorylation state of neuronal proteins, and it is enriched in the striatum and hippocampus [223]. Stimulation of NMDA receptors activates calcineurin upon Ca^2+^- and calmodulin-binding [224]. Calcineurin dephosphorylates mHTT promoting its toxic effects [93]. Knock-in of mHTT in striatal cells caused increased vulnerability to NMDAR stimulation and was strongly associated with elevated calcineurin activity leading to the selective loss of HTT phosphorylation at Ser-421 and contributing to neuronal cell death in HD [225].

Palmitoylation is another PTM that regulates the trafficking of glutamate receptors (GluR) characterized by the covalent and reversible linkage of a palmitic acid molecule to a cysteine residue. The goal of this modification is to stabilize proteins in the plasma membrane and control protein shuttling between intracellular compartments by increasing the hydrophobicity of the protein, and therefore, facilitating the interaction with cellular membranes [226]. The palmitoylation of GluN2A and GluN2B subunits of NMDAR occurs at two different positions of the proteins. The regulation of NMDARs by palmitoylation requires the tyrosine phosphorylation of both GluN2A and GluN2B and the palmitoylation of other synaptic NMDAR-interacting proteins, such as PSD-95 [226]. mHTT reduces the palmitoylation of PSD-95, resulting in mislocalization. Moreover, in HD mouse models, the decreased palmitoylation of GluN2B is the result of reduced HIP14 L function. The reduced activity of HIP14 L results in an increase of extrasynaptic surface expression in striatal neurons, leading to increased neuronal susceptibility to NMDA-induced apoptosis and enhanced NMDA excitotoxicity in early-stage HD [46]. The process of palmitoylation is reversible, and the depalmitoylated NMDA facilitates synaptic maturation and prevents excitotoxicity [227].

A key regulator of the pathology of MSNs in HD is DARPP-32. DARPP-32 is the marker for striatal MSNs and is the primary mediator of dopaminergic and multiple ligand signaling in these cells. Dopamine receptor activation triggers the phosphorylation of DARPP-32 at Thr-34 by PKA or PKG and turns DARPP-32 into a PP1 inhibitor. DARPP-32 is also phosphorylated by Cdk5, converting DARPP-32 into a PKA inhibitor. Thr-34-phosphorylated DARPP-32 is dephosphorylated and inactivated by calcineurin and PP2A [47]. Thus, DARPP-32 is a unique dual-function protein, which is critically involved in regulating electrophysiological, transcriptional, and behavioral responses [228]. Presymptomatic HD mice have severe deficiencies in dopamine signaling in the striatum. These include selective reductions in both total and phosphorylation levels of DARPP-32 and other dopamine-regulated phosphoprotein markers of MSNs [48]. HD mice also show defects in dopamine-regulated ion channels and the D_1_ dopamine/DARPP-32 signaling cascade. This pathomechanism is related to the enhanced NMDA-induced excitotoxicity that provokes the reduced expression of DARPP-32, leading to increased PP1 activity and decreased phosphorylation of PP1 substrates, including HTT in HD striatum [92]. The loss of HTT phosphorylation at Ser-421 residue through the activation of PP1 contributes to NMDA-induced excitotoxicity and neuronal cell death. [50].

#### 2.5.2. Ion Channels: AMPA, TRPC5

AMPA receptors (AMPAR) are non-NMDA ionotropic glutamate receptors that can initiate cation influx directly upon ligand binding. A lot is known about the PTMs of the cytosolic C-terminal tail of AMPAR subunits (GluA1–4) that modulate the receptor activity in endocytosis, intracellular trafficking, channel conductance, and synaptic plasticity. Still, fewer studies are focused on the HD-related relevance of these modifications. The most widely distributed subunits are GluA1 and A2, and GluA1/2 heteromers constitute the majority of AMPARs [229]. AMPAR dysfunction causes impaired hippocampal synaptic plasticity contributing to cognitive impairment in HD. This defect is related to the signaling pathway of the brain-derived neurotrophic factor, tyrosine receptor kinase B. It contributes to dysregulated AMPAR trafficking by reducing interactions between transmembrane AMPA receptor regulatory proteins and the PDZ-domain scaffold protein, PSD-95. Active, phosphorylated CaMKII, which regulates AMPAR surface diffusion by impacting the interaction between TARP-2 and PSD-95, is reduced in the hippocampus in an HD mouse model [230].

S-palmitoylation of the transient receptor potential channel (TRPC) family of proteins controls the life cycle of the channel [231]. TRPC5 is a Ca^2+^-permeable non-selective cation channel that participates in neurotransmitter release and neurological behavior. TRPC5 can be modified post-translationally at Cys residues of the N-terminal. The S-glutathionylation of Cys-176 and Cys-178 of TRPC5 leads to excess Ca^2+^ influx and causes increased Ca^2+^-dependent apoptosis in the striatum of HD patients [51,52].

#### 2.5.3. Glutamate Homeostasis

The normal maintenance of glutamate homeostasis is also crucial in the central nervous system and several neurodegenerative disorders. The glutamate transporter-1 (GLT-1) or excitatory amino acid transporter 2 (EAAT2) is a Na^+^-dependent transporter, which is predominantly expressed in astrocytes and the presynaptic terminals of excitatory neurons. Although its expression level is low, the transporter is responsible for the removal of excess glutamate from the synaptic cleft and prevents glutamate excitotoxicity [232]. Since downregulation of GLT-1 and a defect in glutamate uptake have been observed in the mouse model of HD [233], this transporter is a potential drug target. Different PTMs regulate the activity and subcellular localization of GLT-1. GLT-1 is palmitoylated, and the decreased palmitoylation of GLT-1 contributes to the defect in glutamate uptake and the enhancement of excitatory transmission in an HD mouse model [53].

### 2.6. Neuroinflammatory Pathways Linked to the Progression of HD

Neuroinflammation is one of the hallmarks of HD and might be associated with other neurodegenerative diseases. Interestingly, the brain-resident and peripheral immune cells appear not to have an impact on the inflammation of the CNS in HD. However, the cell-autonomous microglia reactivity promotes pro-inflammatory gene expression and immune reaction. In contrast to other neurological diseases, such as multiple sclerosis and AD, an influx of peripheral immune cells, including lymphocytes and neutrophils, has not been reported in neuropathological studies in HD. Furthermore, increased T-cells were not detected in post-mortem human HD tissues. The neuroinflammatory profile of HD is also quite distinct from other neurodegenerative disorders [234]. In human HD tissues, the level of inflammatory mediators, such as IL-1β and TNF-α, was increased only in the striatum. IL-6, IL-8, and matrix metalloproteinase-9 (MMP-9) were upregulated in the cortex and the cerebellum. In contrast, in PD and AD, a more generalized neuroinflammatory profile was observed with the upregulation of a wide range of chemokines and cytokines [235,236]. The inflammatory mediators detected in the striatum are a sign of the pathology HD, while the widely dysregulated factors, such as IL-6, IL-8, and MMP-9, reflect the effect of mHTT expression [234,237]. Furthermore, the plasma level of IL-18 is significantly reduced in HD patients, suggesting that the classical inflammasome pathway is not involved in the overall inflammatory process in HD. Overall, this indicates that although neuroinflammation and microglia activation are not the primary cause of HD, microglial mHTT expression triggers the autonomous release of pro-inflammatory cytokines, reactive oxygen species, and neurotoxic metabolites, such as prostaglandins and nitric oxide (NO) [238].

NF-κB is a major downstream transcription factor responsible for promoting the transcription of inflammatory mediators upon stimulus. The abnormal activation of the NF-κB pathway is upregulated by mHTT contributing to neurotoxicity in HD patients. mHTT binds to the IκB kinase (IKK) γ subunit of the IKK complex. This promotes the assembly and activation of the IKK complex, which contains the IKKα and IKKβ subunits. The IKKβ kinase phosphorylates IκBα leading to its polyubiquitination and proteasomal degradation and causing the liberation of NF-κB from the complex. Liberated NF-κB translocates to the nucleus and activates transcription of pro-inflammatory cytokines [239,240]. Increased activity of IKK correlates with increased PP2A activity, which dephosphorylates Ser-421 of HTT, thus increasing its nuclear accumulation [241]. IKKβ is degraded more rapidly in patients with HD when compared to controls due to IKK activation. Phosphorylated IKKβ is also increased in monocytes isolated from patients with HD compared to control subjects [242]. IKKβ phosphorylation of HTT at Ser-13 and Ser-16 induces HTT clearance, reduces aggregate accumulation, and blocks HD progression in HD mice, but this process becomes impaired by the polyQ expansion of HTT. The cytokine-induced inflammatory IKK activates the phosphorylation of HTT at Ser-13 and Ser-16, enhancing the removal of HTT by proteasomal and lysosomal degradation. Therefore, the inflammatory IKKβ subunit is required in vivo to slow the early stages of HD progression and behavioral progression in the mouse model and to suppress neurodegeneration and microglial activation leading to neuroinflammation in HD [81].

The NF-κB signaling cascade acts in parallel with other pathways related to neuroinflammation, including the signaling pathways initiated by PI3K, Akt, and MAPKs (including p38, JNK, and ERK1/2) [243,244]. The Akt-related pro-survival pathway is significantly altered in HD and is implicated as a key signal transduction process regulating the toxicity of mHTT. In HD brain, Akt may have a neuroprotective effect via attenuating mHTT toxicity by phosphorylating mHTT at Ser-421. The protein expression level of Akt is increased while the amount of Ser-473 phosphorylated is unchanged. Therefore, activated Akt is unchanged in lymphocytes from HD patients, suggesting that the ratio of activated Akt is much lower in HD patients compared to controls [39].

PI3K activates Akt via phosphorylation at Thr-308 and Ser-473 and these phosphoresidues are dephosphorylated by PP2A and the PH domain leucine-rich repeat protein phosphatase (PHLPP), respectively [245]. PHLPP protein levels are decreased in human HD brains parallel to increased Akt phosphorylation levels at Ser-473, contributing to the high level of activated Akt, the delay in cell death, and the recovery of neuronal viability [40]. Akt levels are decreased in the striatum of patients with HD and, in parallel, the caspase-3-cleaved 49 kDa Akt product has been observed in the cerebellum and the cortex of HD patients [87], suggesting that the caspase-3 protease is activated in patients with HD [39].

Degenerated neurons, activated microglia, pericytes, and astrocytes are characterized by the activation of the Janus kinase/signal transducer and activator of transcription (JAK/STAT) pathway [225] in HD. STAT3 accumulates in the nucleus of reactive astrocytes in the striatum of mouse and primate HD models. However, *Trager* et al. found elevated phosphorylation of STAT5 only in HD gene carriers’ monocytes [246].

Other candidates for the regulation of HD neuroinflammatory processes are the toll-like receptors (TLR), a family of innate immune receptors, with pleiotropic effects on neuronal plasticity and neurodevelopment. TLRs are present in the brain with the strongest expression in microglial and glial cells [247]. TLR 2, 3, and 4 deficiency significantly extends the life expectancy of HD mice [248]. TLRs are expressed in the microglia and glial cells of HD patients and they are triggered by the Danger Associated Molecular Patterns (DAMPs) released by dying neurons upon tissue injury or inflammation [249]. DAMPs activate the innate immune receptors, including TLRs, which results in the translocation of NF-κB to the nucleus and the activation of pro-inflammatory molecule transcription, including IL-6, IL-8, and TNF-α, leading to neuroinflammation [248].

The MAPK pathway is activated in many cell types in patients and mouse models with neurodegenerative conditions. Still, surprisingly there is no evidence showing that it is directly involved in the initiation of astrocyte reactivity and activation of microglia [225,250]. The negative regulator of the MAPK phosphorylation cascade, MAPK phosphatase (MKP-1 or DUSP1), is associated with the pathomechanism of HD and has a neuroprotective role via inhibition of the JNK/p38 pathway [41]. Recent studies revealed the downregulation of the MKP-1 gene (*mkp*1) in human post-mortem HD brain samples and a lack of MKP-1 eventuated neurodegeneration [42].

### 2.7. Transcriptional Dysregulation and Related PTMs in HD

Transcriptional dysregulation is an early problem in HD and includes alterations in both histone modifications and the function of gene expression regulators. These complications have been extensively researched, and the key findings are summarized by several review articles [56,251,252]. Wild-type HTT interacts with several transcription factors (CREB, SP1, NF-κB, NeuroD, p53, UBF), transcriptional activators and repressors (TAFII130, CA150, NCOR, REST/NRSF, PGC-1α), and nuclear receptors (LXRα, PPARγ, VDR, TRα1) (reviewed in [251,252]). Through these interactions, HTT can modulate the transcription of numerous genes. In HD, the function of these transcriptional regulators is disrupted either by loss of interaction with wtHTT or abnormal interaction with mHTT. In the present review, we focus on selected examples of malfunctions of gene expression regulation, where PTMs are also at play.

Both histone acetylation and methylation are altered in HD [56]. Histone acetylation marks are reduced in gene loci relevant to HD, presumably caused by an imbalance in histone acyltransferase/deacetylase activity [253]. The histone acetyltransferase, CREB binding protein (CBP), is sequestered by mHTT. This depletion of soluble CBP results in histone hypoacetylation leading to altered gene transcription that contributes to mHTT-induced neurotoxicity [254,255]. In the hippocampus of HD knock-in mice, reduced levels of CBP and histone H3 acetylation correlate with the downregulation of CREB/CBP-dependent genes responsible for synaptic plasticity and long-term memory [256]. *Gao* et al. demonstrated that wtHTT and CBP are part of a large protein complex (the so-called TCR) that regulates DNA repair during transcriptional elongation. However, mHTT impairs the enzymatic activity of the component proteins, and hence, the function of the TCR [59]. The deubiquitinating enzyme ataxin-3 (ATXN3) is also part of the TCR, and its inactivation by mHTT favors CBP ubiquitination and subsequent degradation. Histone deacetylation is mediated by histone deacetylases (HDACs) and sirtuin deacetylases (SIRTs). Chemical inhibition of HDACs reduces HD-related phenotypes. However, genetic depletion of various HDAC isoforms is rather ineffective in ameliorating neurodegeneration in different HD mouse models [257].

Histone lysine methylation is also affected in HD. For instance, the expression of the histone methyltransferase, ESET (ERG-associated protein with SET domain), and the level of trimethylated histone H3 (K9) is increased in HD patients [258]. Systematic genetic interaction studies in a *Drosophila melanogaster* HD model revealed that reduction of the histone demethylase, Utx, can significantly attenuate mHTT-induced pathology [259].

Normal HTT interacts with protein arginine methyltransferase 5 (PRMT5) and increases its activity towards histones in vitro [260]. However, in HD, PRMT5 function is impaired, and the symmetric arginine dimethylation of histones is reduced, suggesting that PRMT5-mHTT interaction may contribute to the failures of gene expression regulation in HD.

Finally, the accumulation of ubiquitin conjugates in polyQ aggregates depletes nuclear ubiquitin and results in the deubiquitination of histones, thereby reducing the capacity of the cell to perform DNA repair. Since neurons are not replaceable, their longevity heavily relies on a coordinated response to DNA damage and repair. Thus, perturbation of ubiquitin homeostasis likely leads to cell aging through compromising the DNA damage response [261].

Less is known about the contribution of aberrant PTMs in transcriptional dysregulation in HD. CyclicAMP response element-binding protein (CREB) is a ubiquitously expressed transcription factor. Activation of CREB requires phosphorylation at Ser-133 and subsequent binding to CBP, which functions as a transcriptional co-activator [262]. Phosphorylation of CREB is reduced, and the CREB pathway is dysregulated in HD [57], which may contribute to the observed repression of brain-derived neurotrophic factor (BDNF) [58]. Systematic administration of recombinant BDNF improves HD-associated deficits in the R6/2 mouse model and correlates with increased levels of phospho-CREB in the hippocampal region [263]. The natural compound, β-lapachone, has beneficial effects on the HD phenotype, and it restores CREB phosphorylation and increases Sirt1 expression and deacetylation of peroxisome proliferator-activated receptor gamma coactivator-1α (PGC-1α) by a yet unknown mechanism [60]. Indeed, PGC-1α expression and activity are also impaired in HD, which plays a role in mitochondrial biogenesis [61].

Higher levels of the tumor suppressor p53 were detected in HD-affected brain regions and cells overexpressing mHTT [62,63,264]. PTMs of p53 are also dysregulated in HD. mHTT increases the phosphorylation of p53 at Ser-46, leading to the activation of the p53 pathway and the upregulation of apoptosis-related target genes [63]. mHTT overexpression in HEK 293 cells results in elevated p53 phosphorylation at Ser-15, decreased Lys-382 acetylation, and altered p53 ubiquitination. This amplifies p53-mediated transcription and promoting cell death and neurodegeneration [62]. These suggest that inhibition of p53 phosphorylation might play a therapeutic role in HD.

Ribosomal DNA (rDNA) transcription is also reduced in HD. The nucleolar transcription factor, UBF (upstream binding factor) has an essential role in maintaining rDNA transcription. Increased trimethylation of UBF by the histone methyltransferase ESET leads to repressed rDNA transcription in HD [64]. Loss of CBP-mediated acetylation of UBF is another cause of decreased ribosomal RNA expression [65].

Translation of the mRNA of mHTT is regulated by phosphorylation and ubiquitination processes. The 40 S ribosomal S6 kinase (S6 K) is a key enzyme initiating the translation of mHTT mRNA. S6 K is activated by mammalian target of rapamycin mTOR kinase (mTOR) and dephosphorylated by PP2A. The microtubule-associated E3 ubiquitin ligase (MID1) associates with the a4 regulatory subunit of PP2A and initiates its proteosomal degradation [66]. This leads to the activation of S6 K and the facilitation of mHTT mRNA translation. MID1 also associates with the CAG-repeat of m*HTT* in a length-dependent manner, increasing the production of mHTT even more [265].

The protein quality control is compartmentalized in eukaryotic cells such as in neurons. The failure of protein quality control could lead to neurodegenerative disorders such as HD. Increased amounts of misfolded proteins initiate the transcription of ER-chaperone encoding genes [266]. The folding capacity of cells is modulated through the phosphorylation of eukaryotic translation initiation factor 2 (eIF2α) by PERK, a transmembrane ER-resident kinase, which reduces the protein flux and favors protein folding. The PP1 catalytic subunit in a complex with PPP1 R15A or R15B regulatory subunits ensures the reversibility of the process by dephosphorylating phospho-eIF2 [67].

## 3. Concluding Remarks

Endless efforts are made to understand the pathological mechanisms of HD better, to develop new, efficient drugs, which prevent the progression of HD, and to assist in early and accurate diagnosis. Recent publications highlight PTMs with therapeutic potentials in HD. Therefore, we first summarized and highlighted the importance of multiple PTMs involved in aggregation formation. Second, considering that modifications of mHTT do not occur in vitro, but under in vivo conditions, we had to include a larger context. Thus, our study also focused on PTMs involved in diverse cellular processes that are affected, initiated, or inhibited by HD. New therapeutic approaches are being developed based on the mapping of PTMs. For example, the development of new proteomic methods to unveil therapeutic targets for HD by identifying enzyme-substrate interactions [267]. Developing new therapies for HD, in which PTMs are specifically targeted, might add a new approach to the treatment of HD. Taking advantages of the well-characterized PTMs and their physiological consequences, a method to treat HD in combination with other therapeutic novelties might be developed. This approach could include genetic reprogramming, manipulation of metabolic pathways to provide neuroprotection, reduction of mHTT level by regulated proteolysis, or down-regulation of mHTT by CRISPR-Cas9 gene editing.

## Figures and Tables

**Figure 1 ijms-21-04282-f001:**
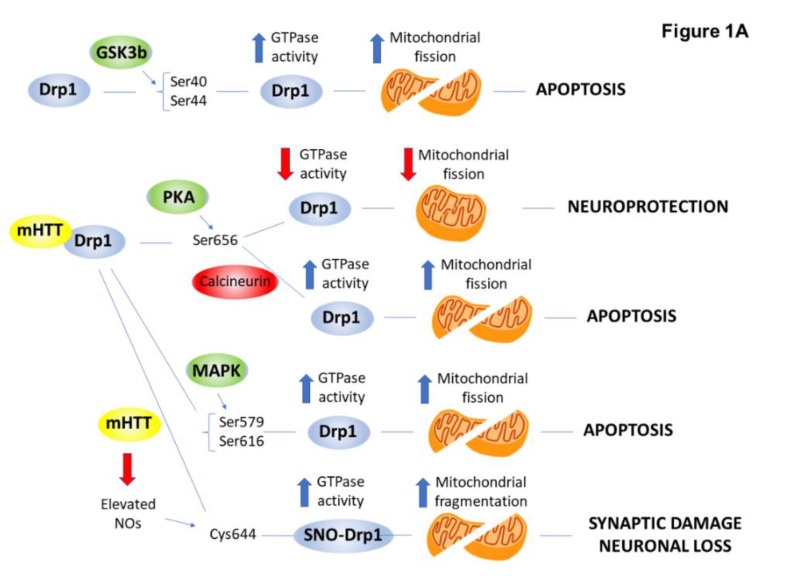
Recently identified post-translational modifications affecting (**A**) mitochondrial dynamics and (**B**) mitochondrial biogenesis in HD. (**A**) Drp1, one of the key players in mitochondrial fission, can be phosphorylated at several Serine amino acid residues leading to imbalanced fission/fusion of mitochondria and determining cell fate in HD. (**B**) Mitochondrial biogenesis is also regulated by the deacetylase activity of the sirtuin SIRT3. SIRT3 prevents cells from mitochondrial dysfunction in HD.

**Figure 2 ijms-21-04282-f002:**
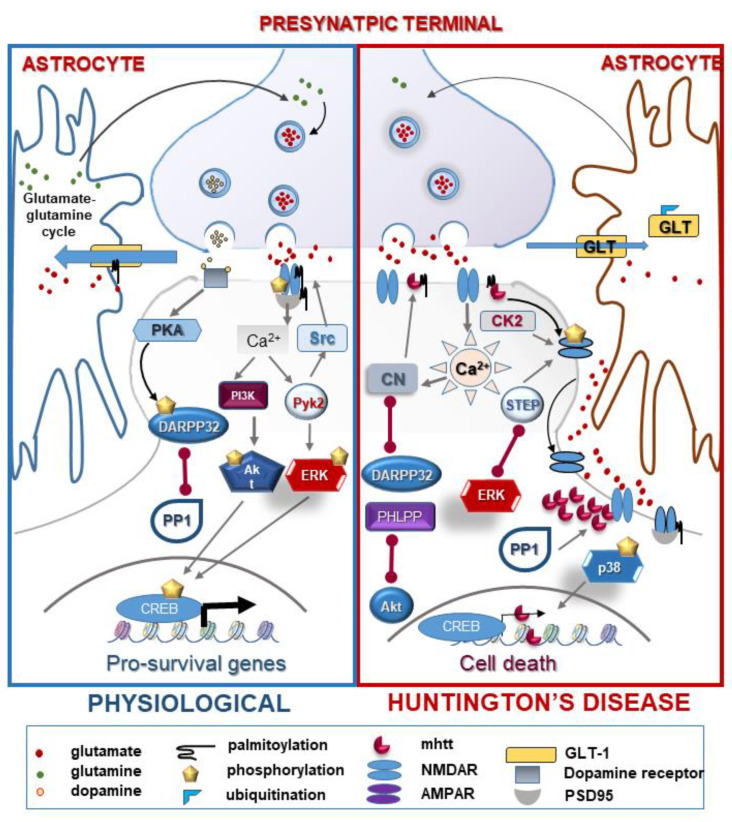
Summary of players in the process of excitotoxicity in HD. The glutamate level increases in the presence of mutant huntingtin. Trafficking of NMDAR and AMPAR from the ER is imbalanced. The activation of extrasynaptic NMDAR is increased, leading to neuronal cell death by the inhibition of ERK, and the activation of the transcription factor CREB. The downregulation of the glial glutamate transporter (GLT-1) results in a defect of its translocation to the plasma membrane. This leads to an increased glutamate level at the synaptic cleft. Arrows represent activation, and red lines, inhibitory processes.

**Table 1 ijms-21-04282-t001:** Post-translational modification of effector proteins in the signal transduction of Huntington’s disease (HD).

Target Protein in HD (Abbreviation)	Modification (Enzymes)	Alteration in HD	Affected Cellular Process	Ref.
**Protein aggregation**				
**Huntingtin (HTT)**	phosphorylation (IKK,CK2,NLK,Akt, SGK,CDK5/PP1,PP2A, PP2B)	↓	mHTT aggregation	[11]
	acetylation (CBP/HDAC1)	↓	formation of fibrillary aggregates, lipid-binding	[11,12]
	ubiquitination	↑/↓	proteosomal degradation	[13]
	SUMOylation (PIAS1, RHES)	↑	escape insoluble aggregate formation, neurotoxicity	[14]
	palmitoylation	↓	inclusion formation	[15]
	myristoylation	↓	pathogenic proteolysis	[16]
	caspase cleavage (caspase-1, -6)	↑	mHTT aggregation	[17,18]
**Ras Homolog Enriched in Striatum (RHES)**	farnesylation	↑	abolished SUMOylation of mHTT	[19,20]
**Proteolytic cleavage**				
**Caspase-6 (CASP6)**	palmitoylation	↑	CASP6 activation	[17]
**Apoptotic protease-activating factor 1 (Apaf-1)**	ubiquitination	↑	regulation of caspase-9	[21]
**Tau impairment and cytoskeletal alterations**				
**Serine/arginine-rich splicing factor-6 (SRSF6, aka SRp55)**	phosphorylation (Dyrk1A)	↑	faulty splicing of tau	[22,23]
**Tau**	phosphorylation (CDK5/PP2B)	↑	tau aggregation	[24]
	caspase cleavage (caspase-2)	↑	tau truncation	[25]
**Tubulin**	acetylation	↓	vesicular transport deficit	[26,27]
**Microtubule-associated protein 1 S (MAPS1)**	acetylation	↓	mHTT degradation	[28]
**β-adducin**	phosphorylation (PKA)	↑	dendritic spine destabilization	[29]
**Mitochondrial abnormalities and defects in energy metabolism**				
**Dynamin-related protein (Drp1)**	phosphorylation (GSK-3β, MAPK1, CDK5/PP2B)	↑	mitochondrial fragmentation	[30,31,32]
	S-nitrosylation	↑	mitochondrial fragmentation	[33,34]
**Manganese superoxide dismutase (MnSOD)**	acetylation	↓	mitochondrial biogenesis	[35,36]
**Lon protease**	acetylation	↓	degradation of aconitase	[37]
**β-catenin**	phosphorylation (GSK-3, CK1)	↓	less efficient energy production	[38]
	ubiquitination	↓		[38]
**Neuroinflammatory pathways**				
**Akt**	phosphorylation (PI3K/PHLPP2,PP2A)	↓	activation of apoptotic signaling pathways	[39,40]
**JNK/p38**	phosphorylation (MKP-1/DUSP1)	↑	loss of neuroprotection	[41,42]
**Excitotoxicity**				
**N-methyl D-aspartate receptors (NMDARs)**	phosphorylation	↑/↓	excitotoxicity disorder of NMDAR trafficking and ER transport	[43,44,45]
	palmitoylation	↓	increased extrasynaptical localization and cellular death	[46]
**Postsynaptic density 95 kDa (PSD-95)**	palmitoylation	↓	disorder in neuronal development, faulty localization of PSD-95	[47,48,49]
**Dopamine- and cAMP-regulated phosphoprotein 32 (DARPP32)**	phosphorylation (PKA/PP1,PP2A)	↓	enhanced NMDA-induced excitotoxicity	[48,50]
**TRCP5**	S-palmitoylation	↑	excess Ca^2+^ influx	[51]
	gluthationylation	↑		[52]
**Glutamate transporter-1 (GLT-1)**	palmitoylation	↓	defect in glutamate uptake	[53]
	ubiquitination	↑		[54]
	nitrosylation	↑		[55]
**Transcriptional dysregulation**				
**Histones**	acetylation	↓	altered gene expression	[56]
	Lys methylation	↓/↑		
	Arg methylation	↓		
	ubiquitination	↓		
**cAMP response element-binding protein (CREB)**	phosphorylation	↓	repressed BDNF expression	[57,58]
**CREB binding protein (CBP)**	ubiquitination	↑	CBP degradation, histone hypoacetylation	[59]
**Peroxisome proliferator-activated receptor gamma coactivator-1α (PGC-1α)**	acetylation	↑	mitochondrial dysfunction	[60,61]
**p53 protein**	phosphorylation	↑	upregulation of apoptosis-related genes	[62,63]
	acetylation	↓		
**Upstream binding factor (UBF)**	trimethylation	↑	repressed rDNA transcription	[64,65]
	acetylation	↓		
**Ribosomal S6 kinase (S6 K)**	phosphorylation	↑	*HTT* transcription	[66]
**Eukaryotic translation initiation factor 2 (eIF2 a)**	phosphorylation (PERK/R15A-PP1, R15B-PP1)		protein quality control	[67]

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
