# Peer review of "How Do Post-Translational Modifications Influence the Pathomechanistic Landscape of Huntington’s Disease? A Comprehensive Review"

_ijms, 2020, doi:10.3390/ijms21124282_

Round 1

Reviewer 1 Report

In this paper named “How do post-translational modifications influence the pathomechanistic landscape of Huntington’s disease? A comprehensive review.” the authors mentioned about the connection between polyglutamine repeats of the huntingtin protein (HTT) and Huntingtin’s Disease (HD). Further, they also discussed different types of post-translational modification (PTM) in HTT and upstream and downstream proteins of neuronal signaling pathways and consequently, their implication to the HD. Then, the authors explained its relevance in different aspects of cell regulation from the dysfunction of proteolytic, mitochondrial, and gene expression to its end i.e.; cell death. In my opinion, the manuscript is well written. There are a few typos problems in the manuscript.

Author Response

IJMS-809063

Response to Reviewers

We would like to thank the Reviewers for their constructive and helpful comments which greatly assisted us to improve and revise the manuscript entitled "How do post-translational modifications influence the pathomechanistic landscape of Huntington’s disease? A comprehensive review." by Beata Lontay, Andrea Kiss, Laszlo Virag and Krisztina Tar.

Responses to reviewers:

Reviewer #1:

In this paper named “How do post-translational modifications influence the pathomechanistic landscape of Huntington’s disease? A comprehensive review.” the authors mentioned about the connection between polyglutamine repeats of the huntingtin protein (HTT) and Huntingtin’s Disease (HD). Further, they also discussed different types of post-translational modification (PTM) in HTT and upstream and downstream proteins of neuronal signaling pathways and consequently, their implication to the HD. Then, the authors explained its relevance in different aspects of cell regulation from the dysfunction of proteolytic, mitochondrial, and gene expression to its end i.e.; cell death. In my opinion, the manuscript is well written.

Comment 1. There are a few typos problems in the manuscript.

Response 1. We thank for the comment. We have revised the grammar and the style of the manuscript and we have corrected the typos in the text (see all modification in track changes). Furthermore, we requested regular editing service from IJMS to improve the grammar and phrasing of our revised manuscript.

Reviewer 2 Report

The review article titled "How do post-translational modifications influence the pathomechanistic landscape of Huntington’s disease? " by Tar et al. describes how posttranslational protein modifications on the mutant and wild type huntingtin protein may impact Huntington's disease (HD) pathogenesis. A comprehenisve list of different posttranslational modifications, including phosphorylation, acetylation, ubiquitylation, sumoylation etc. is discussed bit most space is given to phosphorylation.

The text is full of typos and punctuation errors. Careful editing will be required.

The chapter regarding how posttranslational protein modification on the huntingtin protein modulates its misfolding, degradation, and inclusion formation is quite clear and informative. A thorough discussion of the role of polyQ huntingtin misfolding, aggregation, and inclusion formation in HD pathogenesis should be added here.

The rest of the review article is a rather disorganized list of many different signaling and neuronal pathways and how they might be involved in HD. All these different parts are very hard to follow and are not organized by leading questions or hypotheses in the field.

The table is merely a list of different posttranslational modifications that might be relevant to HD

Author Response

IJMS-809063

Response to Reviewers

We would like to thank the Reviewers for their constructive and helpful comments which greatly assisted us to improve and revise the manuscript entitled "How do post-translational modifications influence the pathomechanistic landscape of Huntington’s disease? A comprehensive review." by Beata Lontay, Andrea Kiss, Laszlo Virag and Krisztina Tar.

Responses to reviewers:

Reviewer #2:

The review article titled "How do post-translational modifications influence the pathomechanistic landscape of Huntington’s disease? " by Tar et al. describes how posttranslational protein modifications on the mutant and wild type huntingtin protein may impact Huntington's disease (HD) pathogenesis. A comprehensive list of different posttranslational modifications, including phosphorylation, acetylation, ubiquitination, sumoylation etc. is discussed bit most space is given to phosphorylation.

Comment 1. The text is full of typos and punctuation errors. Careful editing will be required.

Response 1. The style and language of the manuscript have been revised and corrected (see all modifications in line of the text). In addition, we have requested regular editing service from IJMS to improve the grammar and phrasing of our revised manuscript.

Comment 2. The chapters regarding how posttranslational protein modification (PTM) on the huntingtin protein modulates its misfolding, proteasomal degradation, and inclusion formation is quite clear and informative. A thorough discussion of the role of polyQ huntingtin misfolding, aggregation, and inclusion formation in HD pathogenesis should be added here.

Response 2. The Introduction part has been significantly restructured following the suggestion of the Reviewer. More detailed discussion on the role of polyQ huntingtin misfolding has been added. We described the mutation of htt gene as the cause of HD, and we introduced the cellular function of wild type huntingtin protein. Next, we detailed the aggregate formation and aberrant interactions of the mutant huntingtin protein which lead to the cellular pathomechanism of HD. We have also added a paragraph shortly describing the origin and types of PTMs.

PolyQ huntingtin misfolding, aggregation, and inclusion formation in HD pathogenesis is discussed in details in Chapter 2 and its subchapters. Selected cellular events were listed with the main focus on PTMs, however cellular or organelle abnormalities in HD was shortly introduced to better understand the disease. The development of HD disease is progressive. The majority of dysfunctions ofcellular events and the cellular and molecular pathogenesis that we discuss here already occur before the onset of the disease during the presymptomatic phase.

Comment 3. The rest of the review article is a rather disorganized list of many different signaling and neuronal pathways and how they might be involved in HD. All these different parts are very hard to follow and are not organized by leading questions or hypotheses in the field.

Response 3. The manuscript has been reframed and reorganized as follows:

  1. Introduction to Huntington’s disease
  2. Post-translational modifications in selected cellular events of the diverse pathomechanism of Huntington’s disease
    • PTMs in abnormal HTT protein aggregation
    • Disrupted proteolytic pathways: PTMs in abnormal HTT protein degradation
    • Tau impairement and aberrant cytoskeleton in HD
    • PTMs related to mitochondrial abnormalities and defects in energy metabolism in HD
    • Cell death: focus on excitotoxicity
    • Neuroinflammatory pathways linked to the progression of HD
    • Transcriptional dysregulation and related PTMs in HD
  3. Concluding remarks

Chapter 2 describes signaling and neuronal pathways that are disturbed upon mutant HTT protein expression. The subchapters introduce how the expression, aggregate and/or oligomer formation of mHTT alter and perturbate cellular signaling pathways that are mainly regulated by PTMs of target proteins. These alterations by mHTT expression or aberrant binding/interaction eventually lead to neuroinflammation - an emerging field in neurodegenerative diseases -, and cellular death.

Comment 4.The table is merely a list of different posttranslational modifications that might be relevant to HD.

Response 4. In Table 1 we summarized all PTMs of substrates, which serve as direct modulators of HD pathomechanisms. We classified these targets by their pathophysiological function parallel with the chapters of the review to help readers.